# An octopamine receptor confers selective toxicity of amitraz on honeybees and *Varroa* mites

**Lei Guo[1], Xin-yu Fan[1], Xiaomu Qiao[1], Craig Montell[2], Jia Huang[1]***

[1]Ministry of Agriculture Key Laboratory of Molecular Biology of Crop Pathogens and Insects, Institute of Insect Sciences, Zhejiang University, Hangzhou, China; [2]Department of Molecular, Cellular and Developmental Biology, University of California, Santa Barbara, Santa Barbara, United States

**Abstract** The *Varroa destructor* mite is a devastating parasite of *Apis mellifera* honeybees. They can cause colonies to collapse by spreading viruses and feeding on the fat reserves of adults and larvae. Amitraz is used to control mites due to its low toxicity to bees; however, the mechanism of bee resistance to amitraz remains unknown. In this study, we found that amitraz and its major metabolite potently activated all four mite octopamine receptors. Behavioral assays using *Drosophila* null mutants of octopamine receptors identified one receptor subtype Octβ2R as the sole target of amitraz in vivo. We found that thermogenetic activation of *octβ2R*-expressing neurons mimics amitraz poisoning symptoms in target pests. We next confirmed that the mite Octβ2R was more sensitive to amitraz and its metabolite than the bee Octβ2R in pharmacological assays and transgenic flies. Furthermore, replacement of three bee-specific residues with the counterparts in the mite receptor increased amitraz sensitivity of the bee Octβ2R, indicating that the relative insensitivity of their receptor is the major mechanism for honeybees to resist amitraz. The present findings have important implications for resistance management and the design of safer insecticides that selectively target pests while maintaining low toxicity to non-target pollinators.

**\*For correspondence:**
huangj@zju.edu.cn

**Competing interest:** The authors declare that no competing interests exist.

## Introduction

The western honeybee, *Apis mellifera*, is the most common and important pollinator. They visit flowering plants to collect nectar and move pollen between flowers. However, large colony losses have been reported worldwide over the last few decades in large part due to a syndrome called colony collapse disorder (*Ratnieks and Carreck, 2010*; *Tylianakis, 2013*). Factors that appear to contribute to colony collapse disorder and global bee decline include land-use intensification, pesticides (*Martin et al., 2012*; *Woodcock et al., 2017*), and the parasitic mite *Varroa destructor*, which transmits pathogens (*Stokstad, 2019*; *Wilfert et al., 2016*).

*Varroa* mites colonize beehives and are the most destructive pest of managed honeybee colonies. Commercial beekeepers rely on chemical control agents against *Varroa* mites. However, it is quite difficult to selectively kill mites without impacting on the insect hosts. Currently, only a few synthetic insecticides/acaricides are employed for this purpose such as tau-fluvalinate, coumaphos, and amitraz (*Johnson et al., 2013*). Excessive dependence on tau-fluvalinate and coumaphos has led to high levels of resistance and loss of effectiveness (*González-Cabrera et al., 2016*; *Higes et al., 2020*). In contrast, wide-ranging amitraz resistance is rare. Consequently, amitraz is still effective in controlling *Varroa* populations in commercial beekeeping operations (*Kamler et al., 2016*; *Rinkevich, 2020*).

Many commonly used insecticides, such as neonicotinoids and pyrethroids, are safe for mammals. However, most of them are toxic to bees since they act on ion channels and receptors that are usually

highly conserved in insects. Therefore, it is important to find out why tau-fluvalinate, coumaphos, and amitraz are all effective for *Varroa* mites but have low toxicity to honeybees. Previous studies found that the P450 inhibitor piperonyl butoxide elevated the toxicity of tau-fluvalinate and coumaphos in bees (*Johnson et al., 2006*; *Johnson et al., 2009*) by inhibiting three P450 enzymes that belong to the CYP9Q family (*Mao et al., 2011*). On the other hand, no metabolite enzyme inhibitors enhance the toxicity of amitraz to honeybees (*Johnson et al., 2013*), indicating that pharmacological differences of the molecular target rather than metabolic detoxication may account for the bee's resistance to amitraz.

Amitraz is particularly effective against mites and ticks, as well as first instar larvae of lepidoptera (*Hollingworth, 1976*). Like other formamidines, amitraz can mimic the actions of octopamine (OA), the invertebrate analog of norepinephrine (*Evans and Gee, 1980*; *Hollingworth and Murdock, 1980*). It acts as a high-affinity agonist of OA receptors, which belong to the family of rhodopsin-like G protein coupled receptors (GPCRs). In invertebrates, there are at least four classes of OA receptors: Octα1R (α1-adrenergic-like octopamine receptor, also referred to as OAMB or OA1), Octα2R (α2-adrenergic-like octopamine receptor, also referred to as OA3), OctβR (β-adrenergic-like octopamine receptor, also referred to as OA2), and Oct-TyrR (octopamine/tyramine receptor, also referred to as TAR1) (*Qi et al., 2017*). Thus, it is not known which one is the molecular target of amitraz in vivo. Amitraz is also used to control the cattle tick, *Rhipicephalus microplus*, and point mutations in the Octβ2R or Oct-TyrR were found in resistant strains (*Baron et al., 2015*; *Chen et al., 2007*; *Corley et al., 2013*), suggesting that modification of these receptors could be the underlying mechanism.

In this study, we identified four *Varroa* OA receptors that are all activated by amitraz and its main metabolite, $N^2$-(2,4-Dimethylphenyl)-$N^1$-methylformamidine (DPMF), in the nanomolar range. We then used two behavioral assays to screen six *Drosophila melanogaster* OA receptor mutants and found that only *Octβ2R* knockout flies were insensitive to amitraz. Artificial activation of *octβ2R* neurons also induced amitraz-like poisoning symptoms. Therefore, Octβ2R is the sole molecular target of amitraz in vivo. We further confirmed that VdOctβ2R from *Varroa* mites was more sensitive to amitraz and DPMF than AmOctβ2R from honeybees. Homology modeling and sequence alignment uncovered three amino acids in the predicted binding site of Octβ2R, which confers the selective toxicity of amitraz in both pharmacological assays and transgenic flies. Our results reveal the molecular target of amitraz in vivo and key residues involved in the selectivity of amitraz between *Varroa* mites and honeybees. We suggest that our findings will be very useful for resistance management and the design of bee-friendly insecticides.

## Results

### Amitraz and DPMF can activate multiple *Varroa* octopamine receptors

We identified four candidate *Varroa* OA receptors, which have amino acid homology to the *Drosophila* counterparts VdOAMB (33% identical; 81% coverage), VdOctα2R (61% identical; 44% coverage), VdOct-TyrR (56% identical; 52% coverage), and VdOctβ2R (56% identical; 63% coverage) (*Figure 1A*). Therefore, we tested whether these receptors are sensitive to amitraz and DPMF in a heterologous expression system. When expressed in HEK293 cells, all receptors were functional and showed dose-dependent responses to OA (*Figure 1—figure supplement 1*). Dose-response analysis showed that VdOctβ2R ($EC_{50}$ = 72.8 nM) and VdOctα2R ($EC_{50}$ = 90.2 nM) were more sensitive to amitraz compared to VdOAMB ($EC_{50}$ = 499.5 nM) and VdOct-TyrR ($EC_{50}$ = 548.2 nM) (*Figure 1C*). However, VdOctβ2R ($EC_{50}$ = 6.9 nM) and VdOAMB ($EC_{50}$ = 5.2 nM) were more sensitive to DPMF compared to VdOct-TyrR ($EC_{50}$ = 49.3 nM) and VdOctα2R ($EC_{50}$ = 190.2 nM) (*Figure 1D*). Therefore, we could not exclude any of these proteins as amitraz or DPMF receptors since they all were activated in the nanomolar range. In addition, although three out of four of these *Varroa* receptors showed higher potency and efficacy to DPMF than amitraz, both chemicals were very effective.

An I61F amino acid substitution in the Octβ2R of the cattle tick, *R. microplus*, is associated with amitraz resistance (*Corley et al., 2013*). We expressed the wild-type and I61F mutant of RmOctβ2R in HEK293 cells, but they were not functional (*Figure 1—figure supplement 2*). We found that this residue in transmembrane domain 1 (TM1) is conserved in all representative invertebrate species (*Figure 1B*). Therefore, we generated a I40F (equivalent to I61F in RmOctβ2R) mutant of VdOctβ2R and tested whether the point mutation caused insensitivity to amitraz. However, there was no

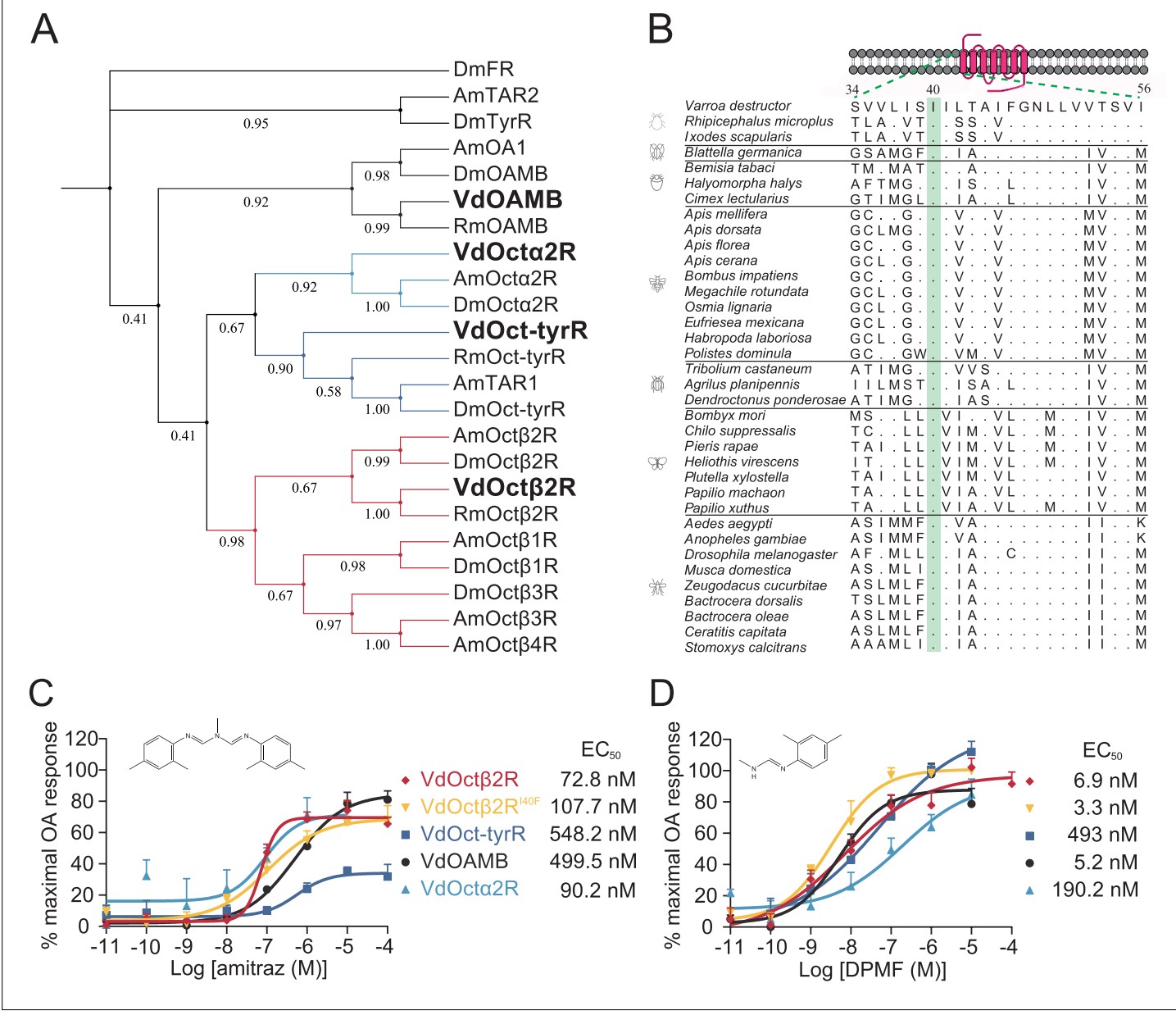

**Figure 1.** Amitraz and its main metabolite DPMF can activate *Varroa* multiple octopamine (OA) receptors in vitro. (**A**) Phylogenetic tree of OA receptors from *Varroa destructor*, *Apis mellifera*, *Rhipicephalus microplus*, and *Drosophila melanogaster*. The values on the branches represent the bootstrap support. The candidate *Varroa* receptors are in bold. (**B**) The Isoleucine40 in the TM1 of Octβ2R, which is associated with amitraz resistance in the cattle tick *Rhipicephalus microplus*, is highly conserved in Arachnida and Insecta. (**C, D**) Dose-response curves of amitraz (**C**) and DPMF (**D**) against the *Varroa* OA receptors. $EC_{50}$ values were calculated using log(agonist) versus response nonlinear fit, mean ± SEM, n = 3 trials, three replicates per trial.

The online version of this article includes the following source data and figure supplement(s) for figure 1:

**Source data 1.** Source data for *Figure 1* and *Figure 1—figure supplement 1* and *Figure 1—figure supplement 2*.

**Figure supplement 1.** Dose-response curves of octopamine (OA) against the indicated OA receptors.

**Figure supplement 2.** Effects of octopamine (OA) and amitraz on the *Rhipicephalus microplus* Octβ2R expressed in HEK293 cells.

significant difference in the $EC_{50}$ between the wild-type and mutant receptors (*Figure 1C and D* and *Figure 1—figure supplement 1*), suggesting that the resistance found in cattle ticks may not be attributed to the I61F mutation in RmOctβ2R. In addition, it is still not clear whether Octβ2R is the molecular target of amitraz.

## Amitraz affects *Drosophila* aggression and locomotion through Octβ2R

Since genetic manipulation of *Varroa* mites has not been established, we took advantage of genetic tools in *Drosophila* to study the mode of action of amitraz, which has been proven useful in elucidating the molecular targets of several insecticides (*Douris et al., 2016*; *Nesterov et al., 2015*). We hypothesized that if amitraz acts on a specific OA receptor in flies, then disruption of the receptor gene would render the mutants insensitive to amitraz. We found that amitraz has low toxicity to wild-type flies as even 5 mM amitraz caused no significant lethality after they were exposed to an amitraz diet for 4 days (*Figure 2—figure supplement 1*). Thus, we used behavioral assays to compare the effects of amitraz on control and mutant files.

OA is both a neuromodulator and neurotransmitter and therefore influences a range of behaviors in insects, including aggression (*Hoyer et al., 2008*), locomotion (*Yang et al., 2015*), sleep (*Deng et al., 2019*), and others (*Kim et al., 2017*). It is reported that treatment with chlorodimeform, another formamidine insecticide and octopaminergic agonist, reduces fighting latency and increases the lunging frequency in socially grouped flies (*Zhou et al., 2008*). We first tested whether feeding with 1 mM amitraz would lead to similar effects in flies (*Figure 2A and B*). In comparison to control flies, treatment of group-housed wide-type Canton-S flies with amitraz resulted in more aggressive behaviors. Males markedly increased their lunging frequency and decreased their fighting latency (*Figure 2C and D*). We then applied amitraz to six OA receptor mutants and found that all but the *octβ2R* null allele *octβ2R^{f05679}* showed robust male-male aggression, as seen in the wide-type animals (*Figure 2C and D*). We also combined *octβ2R^{f05679}* with a corresponding deficiency (Df) that uncovers this gene and found that these males were also not affected by amitraz (*Figure 2C and D*). However, heterozygous control males were still responsive to amitraz (*Figure 2C and D*).

To further confirm that amitraz affects flies through Octβ2R rather than other receptors, we continuously fed flies with amitraz and measured locomotor behavior using an automated monitoring system (*Chiu et al., 2010*). When we added 100 μM or 1 mM amitraz to the diet of agarose-sucrose medium (2% agarose and 5% sucrose), wide-type flies exhibited hyperactivity (*Figure 3A*). Same as above, *octβ2R^{f05679}* flies showed no elevation in locomotor activity, while other receptor mutants exhibited an increase in locomotion upon amitraz treatment (*Figure 3B-G*). Taken together, these results suggest that *octβ2R* mutants showed behavioral resistance to amitraz, and the deletion of any other OA receptor genes had no impact on their sensitivity to amitraz. Therefore, Octβ2R is the only receptor that mediates the effects of amitraz in vivo.

## Hyperactivating *octβ2R*-expressing neurons mimic amitraz poisoning symptoms

The way insects react when they are exposed to formamidines is very unusual. At sublethal doses, these insecticides cause abnormal behaviors like dispersal from plants and detachment of ticks from their host, presumably induced by higher motor activity. At higher doses, this hyperactivity can induce tremors that lead to death (*Evans and Gee, 1980*; *Stone et al., 1974*). Octβ2R is coupled to $G_s$, which typically leads to neuronal activation. We then wondered whether artificial activation of *octβ2R* neurons wound induce amitraz-like poisoning symptoms. Thus, we used the thermosensitive cation channel *Drosophila* TRPA1 (*Hamada et al., 2008*) to acutely hyper-stimulate these neurons. We found that expressing *trpA1* in *octβ2R-Gal4* neurons strongly induced hyperactivity behavior at 32°C and eventually led to paralysis (*Figure 4—video 1* and *Figure 4—video 2*), which is similar to the amitraz-induced behavior phenotype in insects and mites (*Roeder, 2005*). We also expressed *trpA1* in *oamb-Gal4*, *oct-tyrR^{Gal4}*, and *octα2R-Gal4* neurons and found that activation of *oamb-Gal4* and *oct-tyrR^{Gal4}* neurons did not show any behavioral defects (*Figure 4*). Although activation of *octα2R-Gal4* neurons also induced knock-down effects, flies were directly paralyzed without hyperactivity ( , *Figure 4—video 1*). Since Octα2R is a $G_i$-coupled receptor, to decrease cAMP upon activation (*Qi et al., 2017*), which is associated with neuronal silencing, we chose to use *UAS-Shibire^{ts}* (*Kitamoto, 2001*) to inhibit *octα2R* neurons. However, silencing of *octα2R-Gal4* neurons produced a 'stop' behavior rather than hyperactivity or paralysis. The flies exhibited almost no translational or rotational body movement (*Video 1*). Therefore, thermogenetic activation of *octβ2R*-expressing neurons in a short time window phenocopies the action of amitraz in target pests, which demonstrates that pharmacological activation of Octβ2R by amitraz in vivo leads to toxicity and finally to death.

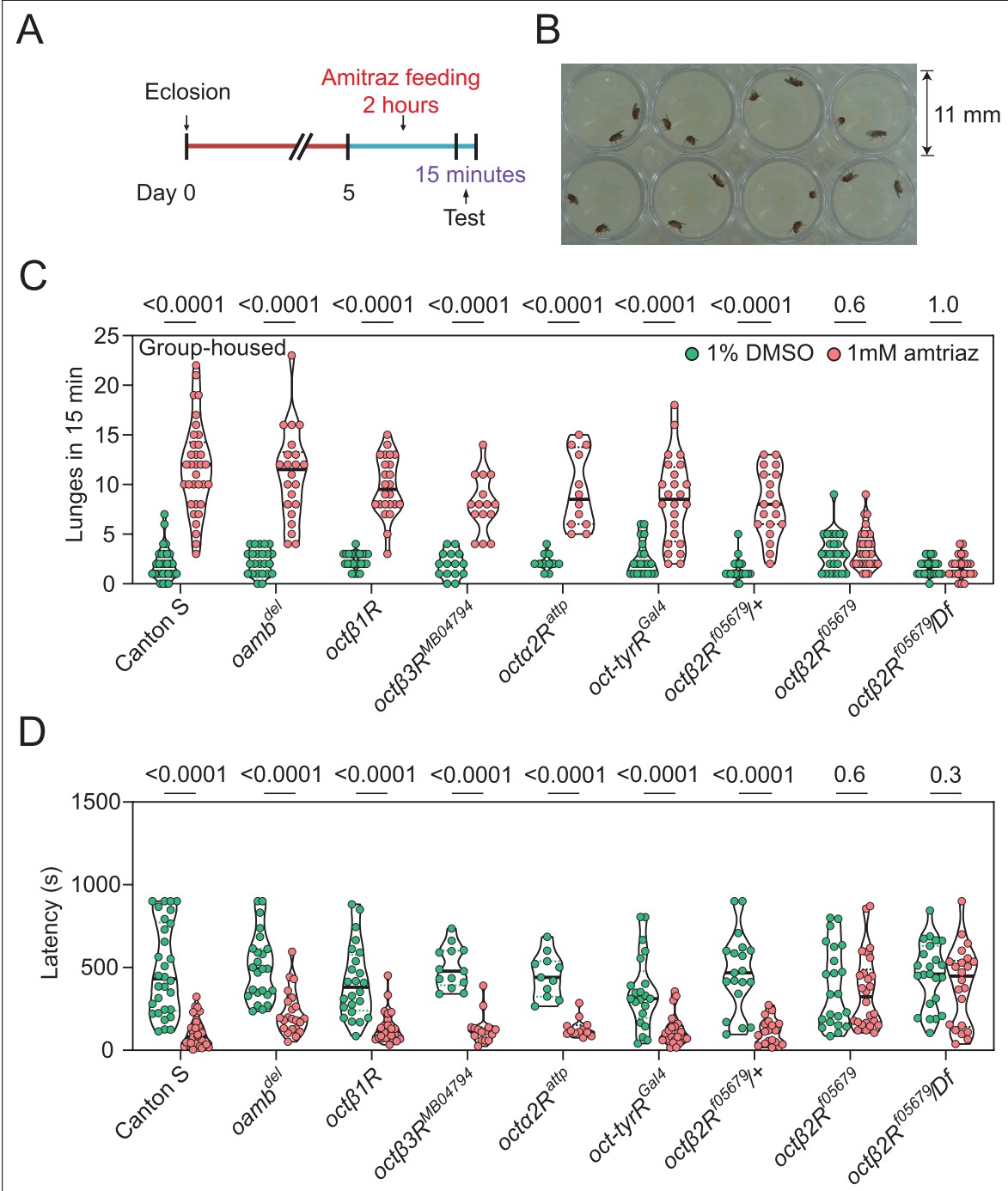

**Figure 2.** Amitraz affects *Drosophila* aggression through Octβ2R. (**A**) Preparation of test flies. In brief, group-housed male flies (~10–15) were fed 1% DMSO plus 5% sucrose (control) or 1 mM amitraz plus 5% sucrose for 2 hr (see Materials and methods). (**B**) The 8-well aggression arena used in this behavioral test. (**C, D**) Effects of 1 mM amitraz on the number of lunges (**C**) and latency to initiate fighting (**D**) in different octopamine (OA) receptor mutants and control flies. p values, Mann–Whitney U tests were performed to analyze statistically significant differences between treatment with 1% DMSO versus 1 mM amitraz in the indicated genotypes, mean ± SEM, n = 12–34.

The online version of this article includes the following source data and figure supplement(s) for figure 2:

**Source data 1.** Source data for *Figure 2* and *Figure 2—figure supplement 1*.

**Figure supplement 1.** Adult survival of flies reared on diets containing 1% DMSO or a range of amitraz concentrations.

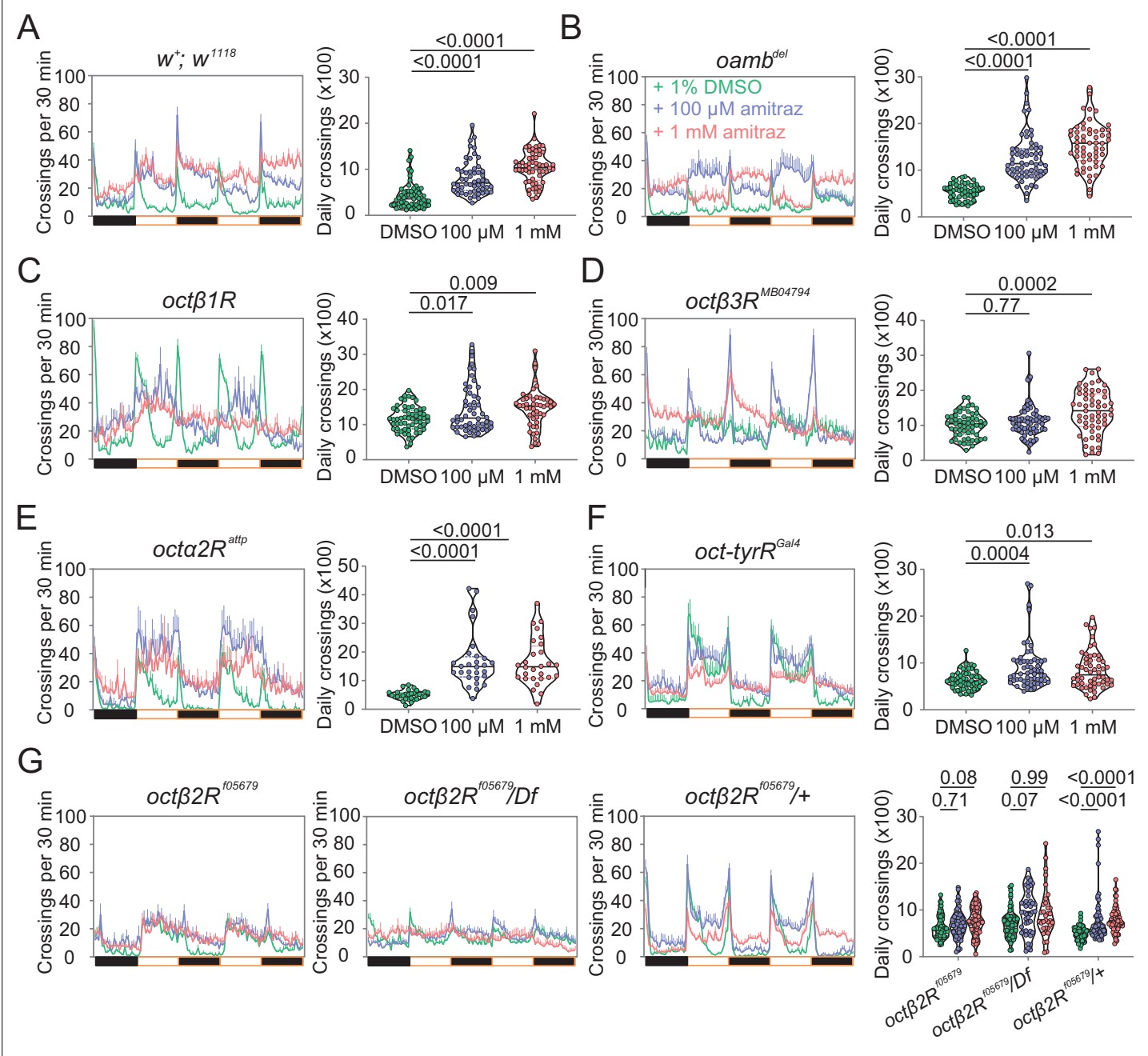

**Figure 3.** Amitraz affects *Drosophila* locomotion through Octβ2R. (**A–G**) Effects of amitraz on midline crossing activity in flies of the indicated genotypes. One 5–7-day-old female fly was gently introduced into each tube, which contained 1% DMSO (control), 100 μM amitraz or 1 mM amitraz added to the agarose-sucrose medium (2% agarose and 5% sucrose) at one end. The other end was sealed with a cotton plug. The tubes were placed in a *Drosophila* Activity Monitor System (see Materials and methods). Black and white bars represent the night and day periods of the 12:12 LD cycle. Yellow boxes indicate the 2-day window of daily crossing activity test. p values, one-way ANOVA and post hoc Bonferroni correction, mean ± SEM, n = 16–32.

The online version of this article includes the following source data for figure 3:

**Source data 1.** Source data for *Figure 3*.

## Honeybee Octβ2R is less sensitive to amitraz than the *Varroa* mite Octβ2R

Previous studies indicated that the differences in receptors may confer the different sensitivities of honeybees and *Varroa* mites to amitraz (*Johnson et al., 2013*). Therefore, we compared the effects of

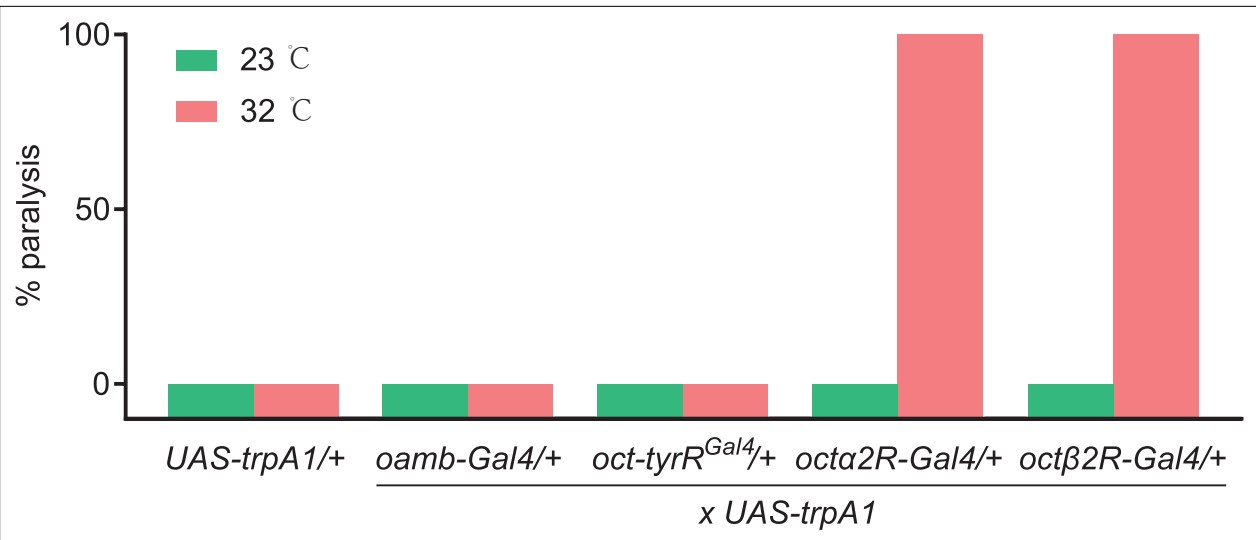

**Figure 4.** The percentage of paralysis behavior, in which four octopamine (OA) receptor-expressing neurons were thermally hyperactivated with *UAS-trpA1*. The following transgenes were used: *oamb-Gal4> UAS-trpA1; oct-tyrR^Gal4> UAS-trpA1; octα2R-Gal4> UAS-trpA1; octβ2R-Gal4> UAS-trpA1.* n = 50–100.

The online version of this article includes the following video and source data for figure 4:

**Source data 1.** Source data for *Figure 4*.

**Figure 4—video 1.** Thermogenetic activation of four octopamine (OA) receptor-expressing neurons using *UAS-trpA1* induces paralysis behavior (related to *Figure 4*).

https://elifesciences.org/articles/68268/figures#fig4video1

**Figure 4—video 2.** Thermogenetic activation of *octβ2R-Gal4* neurons using *UAS-trpA1* induces paralysis behavior (related to *Figure 4*).
https://elifesciences.org/articles/68268/figures#fig4video2

amitraz and DPMF on Octβ2Rs from both species, as well as fruit flies. We found that there were no differences in the AmOctβ2R, VdOctβ2R, and DmOctβ2R OA $EC_{50}$s (*Figure 5—figure supplement 1*). Notably, compared to VdOctβ2R, we found that AmOctβ2R was 16-fold less sensitive to amitraz ($EC_{50}$ = 1.2 μM) and 6-fold less sensitive to DPMF ($EC_{50}$ = 43.4 nM; *Figure 5A and B*).

In the case of DmOctβ2R, it was threefold less sensitive to amitraz ($EC_{50}$ = 242.1 nM), but fivefold more sensitive to DPMF ($EC_{50}$ = 1.4 nM) than VdOctβ2R (*Figure 5A and B*), suggesting that metabolic detoxication may contribute to amitraz tolerance in flies. Actually, we found that both piperonyl butoxide (PBO, a cytochrome P450s inhibitor) and S,S,S-tributylphosphorotrithioate (DEF,

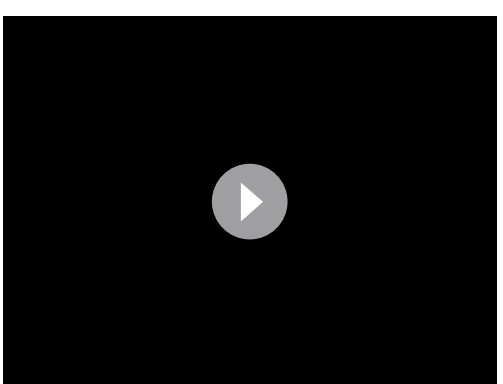

**Video 1.** Silencing of *octα2R-Gal4* neurons using *UAS-Shibire^ts* decreases activity. The following transgenes were used: *octα2R-Gal4> UAS-Shibire^ts*. The movie was speeded up 2×.
https://elifesciences.org/articles/68268/figures#video1

a model carboxylesterase inhibitor) increased the toxicity of amitraz in survival assays. Diethyl maleate (DEM, a model glutathione-S-transferase inhibitor) did not significantly change the toxicity of amitraz (*Figure 5—figure supplement 2*). Thus, we suggest that the fruit fly and honeybee employ different mechanisms to resist amitraz.

We also tested the scenario that amitraz activates multiple OA receptors in *Varroa* mites to cause lethality, but only activates Octβ2Rs in honeybees and flies. Therefore, we cloned the three remaining receptors from fruit flies and honeybees and tested whether these receptors are sensitive to amitraz and DPMF in vitro. Notably, all but DmOAMB showed responses to amitraz and DPMF in nanomolar concentrations (*Figure 5—figure supplement 3*). These results indicate that amitraz and DPMF can bind all

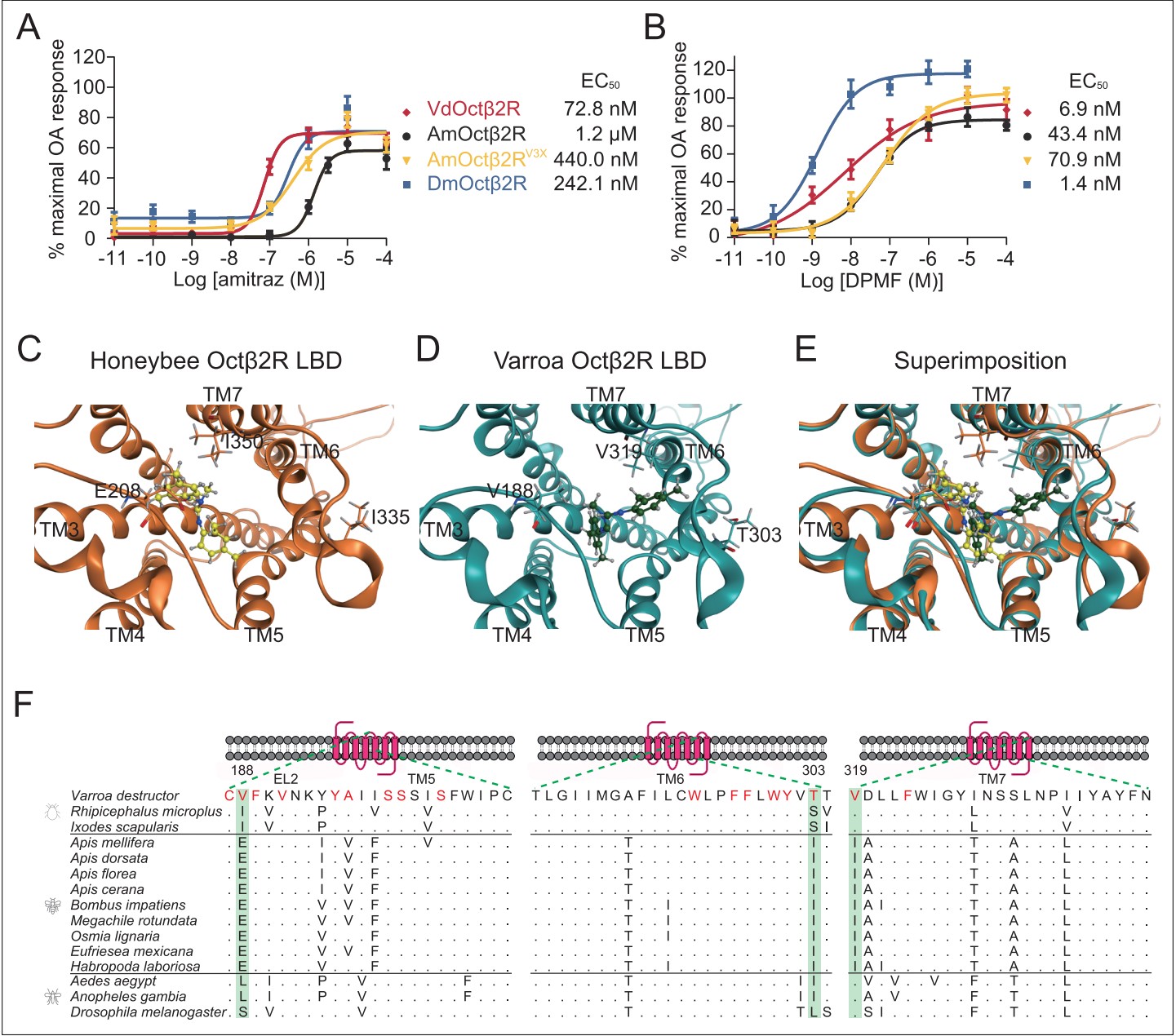

**Figure 5.** Triple amino acids differences determine amitraz sensitivity in Octβ2R in vitro. (**A, B**) The predicted ligand-binding domain of amitraz in the *Varroa* (**A**) and the honeybee (**B**) Octβ2R. Amitraz and three amino acids mutated in this study are shown. (**C**) Superposition of the predicted ligand-binding domain of the honeybee (golden cartoon) and the *Varroa* (blue cartoon) Octβ2R structures. (**D**) Amino acid substitution in the ligand-binding domain (TM5-TM7) of Octβ2R in representative species from Arachnida and Hymenoptera. The predicted amino acids involved in the binding of amitraz are indicated in red. Three amino acids (E208, I335, I350) highlighted in green are conserved among species of bees. EL: extracellular loop; TM: transmembrane domain. (**E, F**) Dose-response curve of amitraz (**E**) and DPMF (**F**) against the indicated octopamine (OA) receptors. EC50 was calculated using log(agonist) versus response nonlinear fit, mean ± SEM, n = 3–4 trials, three replicates per trial.

The online version of this article includes the following source data and figure supplement(s) for figure 5:

**Source data 1.** Source data for *Figure 5* and *Figure 5—figure supplements 1–4*.

**Figure supplement 1.** Dose-response curves of octopamine (OA) against the indicated OA receptors.

**Figure supplement 2.** Adult survival when reared on sucrose solution containing 5 mM amitraz and 1 mM detoxicative enzyme inhibitor.

**Figure supplement 3.** Dose-response curves of OA, amitraz, and DPMF against the fly (**A-C**) and honeybee (**D-F**) OA receptors.

**Figure supplement 4.** Effects of 2,4-dimethylaniline (DMA) and 2,4-dimethylformanilide (DMF) on VdOctβ2R in vitro.

four OA receptors in fruit flies and honeybees, further emphasizing that the relative insensitivity of AmOctβ2R to amitraz is critical for selective toxicity.

## Three residues contribute to resistance of the honeybee Octβ2R to amitraz

We next investigated the molecular mechanism governing the pharmacological differences in Octβ2Rs described above. Amitraz is effective against mites and ticks, but safer for honeybees and for the bumblebee, *Bombus terrestris* (*Marletto et al., 2003*). We reasoned that species-specific sequences of Octβ2Rs, especially around the ligand binding pocket, may determine their sensitivity to amitraz. To identify putative amino acid residues that affect the amitraz response, we generated models of the honeybee and mite Octβ2R by homology modeling using the crystal structure of the carazolol-bond β2-adrenergic receptor (*Huang et al., 2016b*). The models included placement of amitraz into the ligand-binding domain (*Figure 5C-E*). Many residues were predicted to be involved in amitraz binding. However, we found that three amino acids (E208, I335, I350) within the potential ligand-binding domain were unique to bees (*Figure 5F*; highlighted in green). We therefore generated a 'Varroa version' of AmOctβ2R, in which the three amino acids were replaced with corresponding amino acids in VdOctβ2R, and examined its pharmacological profile in cell-based assays. Interestingly, the engineered receptor (AmOctβ2R$^{E208V, I335T, I350V}$, abbreviated to AmOctβ2R$^{V3X}$) was more sensitive (EC$_{50}$ = 440.0 nM) to amitraz than the wide-type AmOctβ2R (EC$_{50}$ = 1.2 μM) (*Figure 5A*), while the OA responses were not affected (*Figure 5—figure supplement 1*). In contrast, AmOctβ2R$^{V3X}$ showed no significant change in DPMF sensitivity (*Figure 5B*). These results indicate that the three residues are responsible for the resistance of honeybees to amitraz.

To test the role of these three amino acid substitutions in vivo, we generated transgenic flies that express AmOctβ2R, AmOctβ2R$^{V3X}$, and VdOctβ2R under the control of the pan-neuronal *elav-GAL4* driver in the *octβ2R* null mutant background. As expected, expression of AmOctβ2R or VdOctβ2R rescued the aggressive and locomotor behaviors induced by amitraz (*Figure 6*). Importantly, 'Varroa flies' showed more aggression and hyperactivity (*Figure 6A and C–F*), while there was no significant effect on the fighting latency (*Figure 6B*). In addition, flies expressing AmOctβ2R$^{V3X}$ displayed a robust increase in the lunging frequency and locomotion than the 'honeybee flies' (*Figure 6A and C–F*). Therefore, these three amino acid changes in honeybee Octβ2R partially phenocopy the amitraz-sensitive properties of the *Varroa* mites.

## Discussion

Although the modes of action of most insecticides are known (https://irac-online.org/), for many the exact molecular targets remain elusive. In order to ascribe whether a candidate protein is indeed the target for an insecticidal effect in vivo, it is not sufficient to demonstrate an in vitro biochemical interaction between an insecticide and a protein. Genetic evidence demonstrating an effect due to mutation of the candidate receptor is critical before it is possible to conclude that a given protein is the target of an insecticide. In some cases, identification of the genetic basis for resistance to a specific insecticide can implicate a putative target (*Douris et al., 2016*; *Van Leeuwen et al., 2012*). A previous study found that a single mutation (I61F) in the cattle tick Octβ2R is associated with amitraz resistance (*Corley et al., 2013*). However, we found that mutation in the equivalent site in *Varroa* Octβ2R did not reduce the potency of amitraz and DPMF. In addition, this isoleucine residue is in TM1 and is unlikely to be involved in agonist binding since the traditional orthosteric site of class A GPCRs is close to a region that includes TM3, TM5, TM6, and TM7 (*Chan et al., 2019*). Thus, more evidence was required to identify the exact molecular target.

Reverse genetics using *D. melanogaster* has been a powerful approach to identify protein targets for insecticides (*Douris et al., 2016*; *Ffrench-Constant et al., 1993*; *Scott and Buchon, 2019*). In cases in which an insecticide is not toxic to the flies, behavioral assays can be employed to characterize potential targets for insecticides. For example, a recent study used climbing assays to identify a *Drosophila* TRPV channel as the target for two insecticides, pymetrozine and pyrifluquinazon (*Nesterov et al., 2015*). In this study, we used two behavioral paradigms to reveal that Octβ2R but not other OA receptors is the molecular target for amitraz. We further showed that transient artificial activation of *octβ2R*-expressing neurons is sufficient to induce amitraz-like poisoning symptoms in

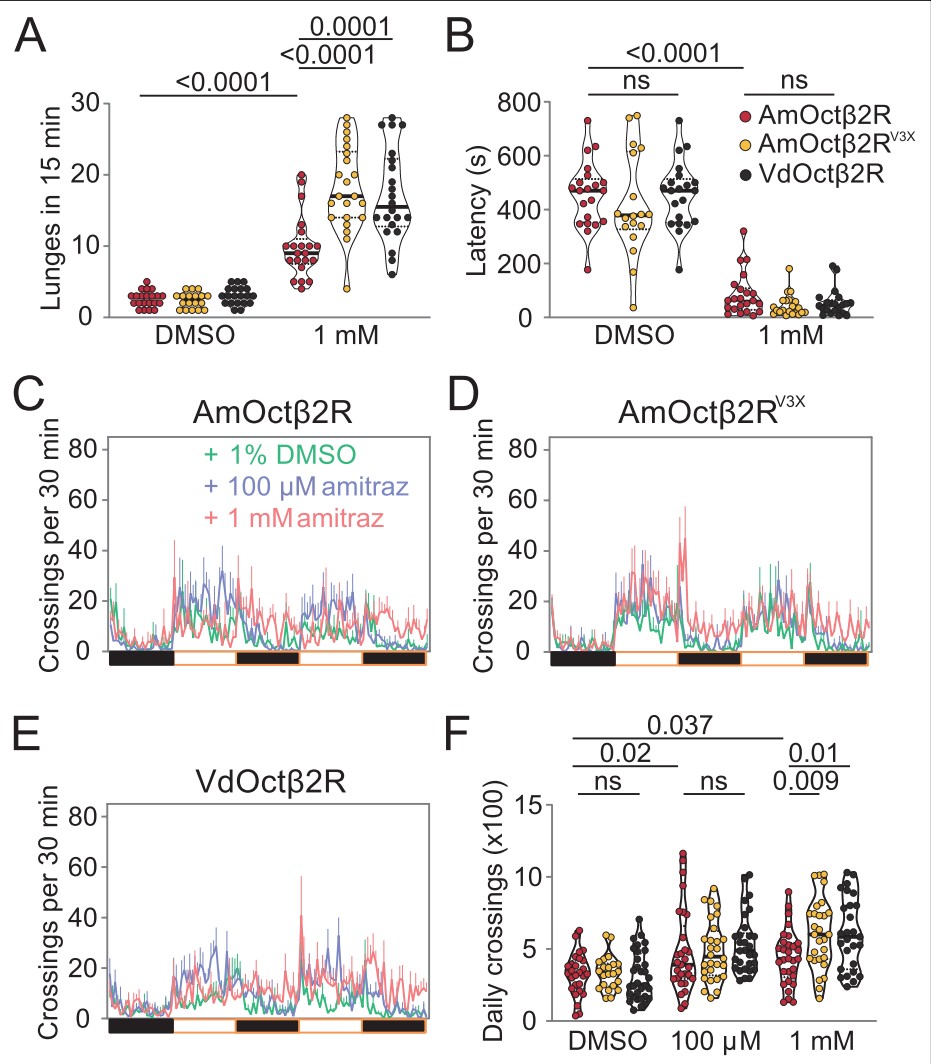

**Figure 6.** Transgenic flies expressing Octβ2R variants show different sensitivities to amitraz. (**A, B**) Number of lunges (**A**) and latencies before initiating fighting (**B**) in the *Octβ2R* null mutant expressing VdOctβ2R, AmOctβ2R, or AmOctβ2R$^{V3X}$. Changes were compared to the AmOctβ2R flies. AmOctβ2R$^{E208V, I335T, I350V}$, abbreviated to AmOctβ2R$^{V3X}$. Genotype: *elav-gal4/UAS-XXOctβ2R; octβ2R$^{f05679}$/octβ2R$^{f05679}$*. p values, Kruskal–Wallis and post hoc Mann–Whitney U tests, mean ± SEM, n = 18–24. (**C–E**) Midline crossing activity in Octβ2R null mutants expressing VdOctβ2R, AmOctβ2R, or the AmOctβ2R$^{V3X}$. (**F**) Daily crossing activities exhibited by *Octβ2R* null mutants expressing VdOctβ2R, AmOctβ2R, or the AmOctβ2R$^{V3X}$. Changes were compared to the AmOctβ2R flies. Genotype: *elav-gal4/UAS-XXOctβ2R; octβ2R$^{f05679}$/octβ2R$^{f05679}$*. p values, two-way ANOVA and post hoc Bonferroni correction, mean ± SEM, n = 16.

The online version of this article includes the following source data for figure 6:

**Source data 1.** Source data for *Figure 6*.

flies. Since Octβ2R is conserved in invertebrates but absent in vertebrates, it could be an ideal target for pest control.

Bees are exposed to a great number of xenobiotics, including plant secondary metabolites that serve as defense compounds against herbivores, various toxins produced by fungi and bacterial, pesticides used in agriculture and other environmental contaminants. Thus, it is not surprising that bees can tolerate some toxic chemicals that occur naturally in their environment. So far, P450-mediated detoxification is the only mechanism known. For instance, the CYP9Q family can not only metabolize the flavonoid, quercetin, but can also detoxify the insecticides coumaphos and tau-fluvalinate (*Mao et al., 2011*), as well as N-cyanoamidine neonicotinoids (*Manjon et al., 2018*). Interestingly,

bumblebees may employ a different mechanism since a voltage-gated sodium channel, which is the molecular target of pyrethroids, is resistant to tau-fluvalinate (*Wu et al., 2017*). Here we found that target-site insensitivity is the major mechanism for bees to resist amitraz. The VdOctβ2R from *Varroa* is more sensitive to amitraz and DPMF than AmOctβ2R from honeybees. Pharmacological and genetic studies revealed that three amino acids replacements increased amitraz sensitivity without a cost to OA sensitivity.

Many insecticides are actually proinsecticides that are transformed into active forms inside insects (*Casida, 2017*). Amitraz has been considered to be a pro-acaricide/insecticide for many years since it can be quickly metabolized to DPMF, 2,4-dimethylformanilide (DMF), 2,4-dimethylaniline (DMA), and others (*Knowles and Hamed, 1989*; *Schuntner and Thompson, 1978*). Thus, it is thought that amitraz undergoes bioactivation in vivo to produce the active metabolite DPMF, although both drugs are active in vivo (*Davenport et al., 1985*). In this study, we found that both amitraz and DPMF are potent Octβ2R agonists while DMF and DMA are not (*Figure 5—figure supplement 4*). DPMF is 11 times more potent than amitraz on VdOctβ2R from *Varroa* mites, which is consistent with a previous report using the silkworm orthologue (*Kita et al., 2016*). On the other hand, DPMF is 28 times more potent than amitraz on AmOctβ2R from honeybees. Amitraz content decreases rapidly to about 1/5 the level of DPMF in ticks and caterpillars (*Knowles and Hamed, 1989*; *Schuntner and Thompson, 1978*). However, it remains at a high level in honeybees even after 24 hr (*Hillier et al., 2013*). Therefore, metabolic differences between mites and bees may further amplify the differential pharmacological effects and contribute to the bee's tolerance to amitraz.

Growing evidence indicates that sublethal doses of insecticides may affect the physiology, cognitive function, and behavior of bees (*Johnson, 2014*). Genetic studies on *Drosophila* have revealed that Octβ2R is involved in behaviors ranging from learning and memory (*Burke et al., 2012*; *Wu et al., 2013*) to ovulation (*Lim et al., 2014*), foraging (*Koon et al., 2011*), and sleep (*Deng et al., 2019*). Further research is needed to examine whether chronic exposure of amitraz affects these bee behaviors negatively. Another major concern is about possible synergism due to exposure of bee colonies to amitraz and other pesticides. Amitraz significantly synergizes the toxic effects of tau-fluvalinate and coumaphos on honeybees (*Johnson et al., 2013*). Amitraz also shows selective synergistic effects for neonicotinoids and pyrethroids against mosquito larvae (*Ahmed and Matsumura, 2012*). Therefore, our identification of the target receptor for amitraz may help clarify the molecular basis of these synergistic effects so that they can be predicted and avoided.

Finally, there is great demand for safer and more selective insecticides that spare beneficial insects. Since the structure and pharmacology of Octβ2R is different between mites and bees, our findings will help target-based screening and the design of novel chemicals acting on this unique molecular target. Some plant-derived essential oils are insecticidal and act on OA receptors (*Jankowska et al., 2018*). Therefore, it will be interesting to test them on *Varroa* mites and Octβ2R.

## Materials and methods

**Key resources table**

| Reagent type (species) or resource | Designation | Source or reference | Identifiers | Additional information |
|---|---|---|---|---|
| Genetic reagent (*Drosophila melanogaster*) | Canton-S | Shanghai Institute of Biochemistry and Cell Biology | Cat#BCF47 | |
| Genetic reagent (*Drosophila melanogaster*) | $w^{1118}$ | Bloomington Drosophila Stock Center | Cat#5905 RRID:BDSC_5905 | |
| Genetic reagent (*Drosophila melanogaster*) | elav-Gal4 | Bloomington Drosophila Stock Center | Cat#8765 RRID:BDSC_8765 | |
| Genetic reagent (*Drosophila melanogaster*) | octβ2R-Df | Bloomington Drosophila Stock Center | Cat#56254 RRID:BDSC_56254 | |

*Continued on next page*

*Continued*

| Reagent type (species) or resource | Designation | Source or reference | Identifiers | Additional information |
|---|---|---|---|---|
| Genetic reagent (*Drosophila melanogaster*) | *octβ3R*[MB04794] | Bloomington Drosophila Stock Center | Cat#24819 RRID:BDSC_24819 | |
| Genetic reagent (*Drosophila melanogaster*) | *octβ1R* | **Koon and Budnik, 2012** | | |
| Genetic reagent (*Drosophila melanogaster*) | *octβ2R*[f05679] | **Lim et al., 2014** | Cat#18896 RRID:BDSC_18896 | |
| Genetic reagent (*Drosophila melanogaster*) | *oamb*[del] | **Deng et al., 2019** | | |
| Genetic reagent (*Drosophila melanogaster*) | *oamb-Gal4* | **Zhou et al., 2012** | | |
| Genetic reagent (*Drosophila melanogaster*) | *octα2R*[attp] | **Deng et al., 2019** | | |
| Genetic reagent (*Drosophila melanogaster*) | *octα2R-Gal4* | **Deng et al., 2019** | | |
| Genetic reagent (*Drosophila melanogaster*) | *octβ2R-Gal4* | **Deng et al., 2019** | | |
| Genetic reagent (*Drosophila melanogaster*) | *UAS-trpA1* | **Hamada et al., 2008** | | |
| Genetic reagent (*Drosophila melanogaster*) | *UAS-Shibire*[ts] | **Kitamoto, 2001** | | |
| Genetic reagent (*Drosophila melanogaster*) | *oct-tyrR*[Gal4] | This paper | | Mutant allele; Materials and methods, 'Fly strains' |
| Genetic reagent (*Drosophila melanogaster*) | *UAS-VdOctβ2R* | This paper | | Mutant allele; Materials and methods, 'Fly strains' |
| Genetic reagent (*Drosophila melanogaster*) | *UAS-AmOctβ2R* | This paper | | Mutant allele; Materials and methods, 'Fly strains' |
| Genetic reagent (*Drosophila melanogaster*) | *UAS-AmOctβ2R*[V3X] | This paper | | Mutant allele; Materials and methods, 'Fly strains' |
| Chemical compound, drug | Amitraz | Sigma | Cat#45323 | |
| Chemical compound, drug | $N^2$-(2,4-Dimethylphenyl)-$N^1$-methyformamidine | Sigma | Cat#BP641 | |
| Chemical compound, drug | 2,4-Dimethylaniline | Sigma | Cat#301493 | |
| Chemical compound, drug | (±)-Octopamine hydrochloride | Sigma | Cat#68631 | |
| Chemical compound, drug | 2,4-Dimethylformanilide | AccuStandard | Cat#P-1100S-CN | |

*Continued on next page*

*Continued*

| Reagent type (species) or resource | Designation | Source or reference | Identifiers | Additional information |
|---|---|---|---|---|
| Chemical compound, drug | Piperonyl butoxide | Aladdin | Cat#P113864 | |
| Chemical compound, drug | S,S,S-Tributylphosphorotrithioate | Aladdin | Cat#T114221 | |
| Chemical compound, drug | Diethyl maleate | Aladdin | Cat#D104017 | |
| Chemical compound, drug | Poly-D-lysine | Sigma | Cat#P0296 | |
| Chemical compound, drug | Sucrose | Sinopharm | Cat#10021418 | |
| Chemical compound, drug | Agarose | Sinopharm | Cat#63005518 | |
| Cell lines | HEK 293 | The Cell Bank of Type Culture Collection of Chinese Academy of Sciences | Cat#GNHu43 | https://www.cellbank.org.cn/ |
| Recombinant DNA reagent | Plasmid: pcDNA3.1-VdOAMB | This paper | | See 'Construction of expression plasmids' |
| Recombinant DNA reagent | Plasmid: pcDNA3.1-VdOctβ2R | This paper | | See 'Construction of expression plasmids' |
| Recombinant DNA reagent | Plasmid: pcDNA3.1-VdOctβ2R$^{I40F}$ | This paper | | See 'Construction of expression plasmids' |
| Recombinant DNA reagent | Plasmid: pcDNA3.1-VdOctα2R | This paper | | See 'Construction of expression plasmids' |
| Recombinant DNA reagent | Plasmid: pcDNA3.1-VdOct-tyrR | This paper | | See 'Construction of expression plasmids' |
| Recombinant DNA reagent | Plasmid: pcDNA3.1-AmOAMB | This paper | | See 'Construction of expression plasmids' |
| Recombinant DNA reagent | Plasmid: pcDNA3.1-AmOctβ2R | This paper | | See 'Construction of expression plasmids' |
| Recombinant DNA reagent | Plasmid: pcDNA3.1-AmOctβ2R$^{V3X}$ | This paper | | See 'Construction of expression plasmids' |
| Recombinant DNA reagent | Plasmid: pcDNA3.1-AmOctα2R | This paper | | See 'Construction of expression plasmids' |
| Recombinant DNA reagent | Plasmid: pcDNA3.1-AmOct-tyrR | This paper | | See 'Construction of expression plasmids' |
| Recombinant DNA reagent | Plasmid: pcDNA3.1-DmOAMB | This paper | | See 'Construction of expression plasmids' |
| Recombinant DNA reagent | Plasmid: pcDNA3.1-DmOctβ2R | This paper | | See 'Construction of expression plasmids' |
| Recombinant DNA reagent | Plasmid: pcDNA3.1-DmOctα2R | This paper | | See 'Construction of expression plasmids' |
| Recombinant DNA reagent | Plasmid: pcDNA3.1-DmOct-tyrR | This paper | | See 'Construction of expression plasmids' |
| Recombinant DNA reagent | Plasmid: pcDNA3.1-RmOctβ2R | This paper | | See 'Construction of expression plasmids' |
| Recombinant DNA reagent | Plasmid: pcDNA3.1-RmOctβ2R$^{I61F}$ | This paper | | See 'Construction of expression plasmids' |
| Software | SoftMax Pro software (v. 7.1.2.0) | Molecular Devices | | https://www.moleculardevices.com/ |

*Continued*

| Reagent type (species) or resource | Designation | Source or reference | Identifiers | Additional information |
|---|---|---|---|---|
| Software | Molecular Operating Environments (MOE, 2015.10) | Chemical Computing Group | | https://www.chemcomp.com/ |
| Software | Prism 7.0 | GraphPad | GraphPad Prism, RRID:SCR_002798 | |
| Other | DMEM media | ThermoFisher Scientific | Cat#10566016 | |
| Other | Lipofectamine 2000 | ThermoFisher Scientific | Cat#11668019 | |
| Other | 96 well polystyrene microplates | ThermoFisher Scientific | Cat#165305 | |
| Other | 0.25% Trypsin-EDTA | ThermoFisher Scientific | Cat#25200072 | |
| Other | Fura 2-AM and Pluronic F-127 | Dojindo Molecular Technologies | Cat#F025 | |

## Fly strains

Flies were maintained and reared on conventional cornmeal-agar-molasses medium at 25 ± 1°C, 60% ± 10% humidity with a photoperiod of 12 hr light:12 hr night (lights on at 7 AM). For experiments using *UAS-trpA1* and *UAS- Shibire^ts* transgenes, flies were reared at 21°C. The Canton-S strain was used as wide-type for aggression assays. For locomotion assays, the $w^{1118}$ strain was crossed to Canton-S so that the X chromosome was $w^+$ while the other chromosomes were from $w^{1118}$ ($w^+$; $w^{1118}$).

The following stains were sourced from the Bloomington Stock Center (Indiana University): *elav-Gal4* (#8765), *octβ2R-Df* (#56254), *octβ3R^MB04794* (#24819), *UAS-trpA1* (**Hamada et al., 2008**), *UAS-Shibire^ts* (**Kitamoto, 2001**), and *octβ2R^f05679* (**Lim et al., 2014**). *octβ1R* was a gift from Dr. Vivian Budnik (**Koon and Budnik, 2012**) (University of Massachusetts Medical School). *oamb^del*, *oamb-Gal4*, *octα2R^attp*, *octα2R-Gal4*, and *octβ2R-Gal4* were gifts from Dr. Yi Rao (**Deng et al., 2019**) (Peking University). *octβ3R^MB04794* has an insertion of a transposable element that disrupts the gene (**Zhang et al., 2013**).

We generated the *oct-tyrR^Gal4* mutant by ends-out homologous recombination, in which the first exon of the gene was replacement by gene encoding the Gal4 transcription activator and the *mini-white* reporter gene as described previously (**Huang et al., 2016a**). We PCR-amplified two 3 kb genomic DNA fragments from isogenic $w^{1118}$, corresponding to the 5′ end of the start codon and the 3′ side of the first exon of the *oct-tyrR* gene, as homologous arms, then subcloned into the pw35Gal4 (**Moon et al., 2009**). The transgenic flies were generated by randomly inserted transgenes, mobilizing transgenes and screening for targeted insertions. The following primers were used for mutant confirmation: *oct-tyrR^Gal4*-F: 5'-CTGTTTGTAAATGTCACCACAACGG-3' *oct-tyrR^Gal4*-R: 5'-CGCCCCAGGATCGAGTAA-3'.

To generate the *UAS-VdOctβ2R* and the *UAS-AmOctβ2R* transgenes, we subcloned the *Varroa* and honeybee *Octβ2R* cDNA sequences into the pUAST-attp vector, respectively. The mutated version of *AmOctβ2R* was directly synthesized by GenScript. The Octβ2R-pUAST construct with $\phi$C31-medidated transgenesis targeted the attp40 specific-site on the second chromosome. The following primers were used for generation of transgenes:

> *UAS-AmOctβ2R*-F: 5'-GGCCGCGGCTCGAGGATGACGACGATCGTGACGAG-3'
> *UAS-AmOctβ2R*-R: 5'-AAAGATCCTCTAGAGTCAGAGGCTGCTACCGTACTCG-3'
> *UAS-AmOctβ2R^V3X*-F: 5'-GGCCGCGGCTCGAGGATGACCACAATCGTGACCAGC-3'
> *UAS-AmOctβ2R^V3X*-R: 5'-AAAGATCCTCTAGAGTCAAAGCTTCAGGCTAGAGCCATACT-3'
> *UAS-VdOctβ2R*-F: 5'-GGCCGCGGCTCGAGGATGTCTGTGGAGGCTGGAGC-3'
> *UAS-VdOctβ2R*-R: 5'-AAAGATCCTCTAGAGTCAAAGCTTTGTCACCAGGGTCTTATATGTAC-3'

## Identification of OA receptors in *V. destructor*

To identify members of the OA receptor gene family in *Varroa*, we performed a two-step analysis: (1) we used the *Drosophila* OA receptor protein sequences as queries to perform BLASTp search against *Varroa* genome (Vdes_3.0) and (2) verified the candidate genes by BLASTp again without a limit of species as previous described (**Guo et al., 2020**). We took all screened genes that were Reciprocal Best Hits with the *Drosophila* OA receptor gene family and then renamed *Varroa* OA receptors.

We next built a phylogenetic tree to show the evolutionary relationships between these OA receptor genes among different species. The *Drosophila* FMRFamide receptor (DmFR) was used as the outgroup. All the amino acid sequences were aligned by Clustal Omega (https://www.ebi.ac.uk/Tools/msa/clustalo/). A neighbor-joining tree was performed by MEGA X with default parameters, 1000 bootstrap replications, and substitution with JTT model (*Kumar et al., 2018*). The screened genes and sequences containing previous accessions are listed in supplement source data.

## Construction of expression plasmids

*Varroa* full-length cDNAs for all predicted OA receptors in genome annotation were cloned from a single mite or synthesized by GenScript. We inserted the Kozak consensus sequence before protein translation initiation site and subcloned with NheI-HindIII sites into the pcDNA3.1(-)-myc-His A vector for expression in mammalian cells.

*Drosophila* and honeybee OA receptors synthesized by GenScript were cloned into the pcDNA3.1(-)-myc-His A vector for expression in mammalian cells. We synthesized a wild-type *R. microplus* β-adrenergic OA receptor (accession number: AFC88978.1) and its I61F mutant.

## Cell culture and transfection

HEK293 cells were obtained from the Cell Bank of Type Culture Collection of Chinese Academy of Sciences (Shanghai, China, Cat#GNHu43), and the identity (STR profiling) and mycoplasma contamination status of this cell line was tested by the supplier (https://www.cellbank.org.cn/search-detail.php?id=770). Cells were grown in DMEM media (ThermoFisher Scientific, Cat#10566016) supplemented with 10% fetal bovine serum according to standard protocol at 37°C and 5% $CO_2$. Cells were seeded on 60 mm Petri dishes until 80–90% confluent at transfection. Cells were transiently transfected, using Lipofectamine 2000 Transfection Reagent (ThermoFisher Scientific, Cat#11668019), according to the manufacturer's protocol, with 2 μg of each OA receptor-expressing plasmid and $G\alpha_{16}$-expressing plasmid (*Offermanns and Simon, 1995*), except OAMB, which is coupled to $G_q$ protein endogenously and can be directly activated to increase intracellular $Ca^{2+}$ levels. The $G\alpha_{16}$ can convert $G_i$/$G_s$-coupled receptors to phospholipase C pathway (*Offermanns and Simon, 1995*).

50 μL of 50 μg/mL poly-D-lysine (Sigma-Aldrich, Cat#P0296) were added to each well of a 96-well polystyrene microplates (ThermoFisher Scientific, Cat#165305) for 4–5 hr and then washed with deionized water for three times. Transfected cells were washed with PBS and dissociated with 200 μL 0.25% Trypsin-EDTA (ThermoFisher Scientific, Cat#25200072) for 30 s at room temperature. Cells were equally seeded into 96-well plates and incubated with DMEM media containing 10% fetal bovine serum for 24–30 hr.

## Calcium mobilization assay

To monitor intracellular $Ca^{2+}$ levels, cells in 96-well microtiter plates were washed three times with 100 μL saline solution containing 152 mM NaCl, 5.4 mM KCl, 0.8 mM $CaCl_2$, 1.8 mM $MgCl_2$, and 5.5 mM glucose, 10 mM HEPES (pH 7.4) (*Qi et al., 2016*), and loaded with 2 μmol/L Fura 2-AM and 0.05% Pluronic F-127 (Dojindo Molecular Technologies, Cat#F025) at 50 μL/well for 40 min in saline solution at 37°C and 5% $CO_2$. After washing three times in saline solution, we pipetted 200 μL of fresh saline solution into each well of a 96-well plate containing the loaded cells and incubated for 15 min at 37°C and 5% $CO_2$.

Calcium mobilization experiments were performed on a FlexStation 3 Multi-Mode Microplate Reader (Molecular Devices). All compounds were prepared freshly 1–2 hr prior to assays at 5× concentration in saline buffer into a 96-well plate. The FlexStation 3 instrument was setup using a SoftMax Pro software according to the recommended experimental protocol. Calcium flux was monitored using the 340/380 nm excitation ratio channel and 510 nm emission wavelength. PMT sensitivity setup medium and three flashes/read. 50 μL of compounds were transferred to each well containing 200 μL of fresh saline solution at 30 s, and fluorescence were recorded every 4 s for 3 min. Normalization of fluorescence data to zero baseline and SoftMax Pro software (v. 7.1.2.0) (Molecular Devices) was used to calculate the ratio of the 340/380 nm wavelengths. Dose-response curve and data analysis were done in GraphPad Prism 7 (GraphPad).

## Bioassay

We introduced ten 5–7-old females into fly vials containing 5 mL 2% (wt/vol) agarose and 5% (wt/vol) sucrose, supplemented with a final amitraz concentration of 0–5 mM. Amitraz concentrations ≥ 10 mM cannot be prepared because of insolubility. The control and each treatment were performed in triplicate. Female mortality rates were monitored for 8 days.

For the enzyme inhibitor assay, we introduced 10 females into a new vial containing sucrose solution supplemented with 5 mM amitraz and 1 mM inhibitor.

## Aggression assay

10–15 newly eclosed male flies were reared for 5–7 days in fly vials containing conventional cornmeal-agar-molasses medium. They were transferred to new vials consisting of filter paper with 300 µL 5% sucrose solution and 1% DMSO (control) or an appropriate concentration of amitraz for 2 hr prior to performing behavior assays. Aggressive behavior was assayed in a '8-well' cylindrical arena, with the dimension of 11 mm and the volume of 1.6 mL. Each arena contained 1 mL of fresh sucrose-agarose food solution (apple juice (Wei Chuan Foods Corporation) consisting of 2.25% (w/v) agarose and 2.5% (w/v) sucrose) (*Asahina et al., 2014*). A drop of yeast solution was added to the center of each well and allowed to air dry. Aggression assays were performed between 7:00–11:00 AM and 4:00–7:00 PM because the flies were most active during these periods (see Locomotion assay). Two male flies were gently introduced into the arenas by aspiration through a small hole in the lid. We then transferred other flies by sliding the lid. Control and drug-treated flies were placed in the arenas at the same time and recorded with a video camera for 15 min. The total number of lunges, which is a stereotyped behavioral pattern in which the winner rears up on its hind legs and uses its front legs snap on the loser, was measured to assess fighting behavior (*Huang et al., 2016a*; *Zhou et al., 2008*). The latency time was the time that elapsed between initiation of the video capture and when the first lunge was completed (*Zhou et al., 2008*).

## Locomotion assay

Locomotor behavior was performed using the *Drosophila* activity monitoring system (DAMS, Trikinetics) as briefly described previously (*Yang et al., 2015*). Individual 5–7-day-old female flies were gently anesthetized and introduced into tubes (5 mm × 65 mm) containing drug medium or no drug medium for 3 days. The tubes were then inserted and fixed firmly in the DAMS. Flies were allowed to adapt to the new environment for 1 day before sampling midline crossing activity every minute and binning data to 30 min. Average daily activities were calculated based on 2 days of testing.

## Thermogenetic activation and silencing assays

Flies for TRPA1-mediated thermogenetic activation and Shibire-mediated silencing experiments were collected upon eclosion and reared in vials containing standard food medium at 21°C for 5–8 days. For thermogenetic activation with the *UAS-trpA1* transgene, 10 flies were transferred to new empty vials by gently inspiration, and then the assays were performed at 23°C and 32°C for 10 min. The percentage of paralysis behavior, in which the animal lies on its back with little effective movement of the legs and wings, was measured. For silencing assays, *UAS-Shibire*[ts] transgene was used and flies were also transferred to fly vials at 23°C and 32°C for 10 min. The 'stop' behavior was defined as a condition that the animal exhibits almost no translational or rotational body movement.

## Molecular docking

Models of the honeybee and *Varroa* Octβ2R-LBD were built using the Molecular Operating Environments (MOE, 2015.10) using the β2-adrenergic receptor and carazolol-bond crystal structures (PDB ID: 5D5A) as homology templates as described (*Hu et al., 2017*). The models were evaluated using Ramachandran plots and the UCLA-DOE server. During molecular docking calculations, water molecules were deleted, 3D protonation was added, and the energy of the protein models was minimized using the MOE algorithm with default parameters. The MOE-dock program was used for docking compounds, and energies were allowed to minimize. To search for the correct conformations during the calculations, the ligands were kept flexible. The default parameters were set according to the rigid receptor docking protocol. Thirty conformations containing the docked poses and scored were

output at the end of the dock operation. A lower binding-free energy in the docking simulation means a better binding interaction between the receptor and the ligand.

## Statistical analysis

All statistical analyses were performed using GraphPad Prism 7 (GraphPad Software). Data for aggressive behavior were analyzed with nonparametric tests. For comparisons of more than three groups, we first used the Kruskal–Wallis test. If the null hypothesis that the median of all experimental groups is the same was rejected, a post hoc Mann–Whitney U test was used for statistical analysis between the relevant pair of genotypes. Data for locomotion were considered to be normally distributed since these data passed the D'Agostino–Pearson omnibus test. Therefore, one-way or two-way ANOVA was performed to test the null hypothesis. If the null hypothesis was rejected, we then used a post hoc Bonferroni correction for multiple comparisons. p values are indicated in the figures. Details of other statistical methods are reported in the figure legends.

## Acknowledgements

We thank Yi Rao (Peking University), Vivian Budnik (University of Massachusetts), Zhefeng Gong, Liming Wang, and Jianhua Huang (Zhejiang University) for fly stocks; Huoqin Zheng (Zhejiang University) for *Varroa* cDNAs; Yinjun Tian (Zhejiang University) for assistance with the locomotion assays; Xueping Hu (Zhejiang University) for assistance with the molecular docking. We also thank Xiao-Wei Wang (Zhejiang University) for helpful comments on the manuscript. Stocks obtained from the Bloomington Drosophila Stock Center (NIH P40OD018537) were used in this study.

## Additional information

### Funding

| Funder | Grant reference number | Author |
| --- | --- | --- |
| National Natural Science Foundation of China | 31572039 | Jia Huang |
| National Natural Science Foundation of China | 32072496 | Jia Huang |
| Zhejiang Provincial Outstanding Youth Science Foundation | LR19C140002 | Jia Huang |
| National Institute on Deafness and Other Communication Disorders | DC007864 | Craig Montell |
| National Institute on Deafness and Other Communication Disorders | DC016278 | Craig Montell |

The funders had no role in study design, data collection and interpretation, or the decision to submit the work for publication.

### Author contributions

Lei Guo, Data curation, Formal analysis, Investigation, Methodology, Visualization, Writing – original draft; Xin-yu Fan, Data curation, Investigation; Xiaomu Qiao, Formal analysis, Investigation; Craig Montell, Formal analysis, Funding acquisition, Resources, Writing – original draft; Jia Huang, Conceptualization, Formal analysis, Funding acquisition, Investigation, Methodology, Project administration, Supervision, Writing – original draft, Writing - review and editing

### Author ORCIDs

Lei Guo ![ORCID] http://orcid.org/0000-0002-7122-3442
Craig Montell ![ORCID] http://orcid.org/0000-0001-5637-1482
Jia Huang ![ORCID] http://orcid.org/0000-0001-8336-1562

Decision letter and Author response
Decision letter https://doi.org/10.7554/eLife.68268.sa1
Author response https://doi.org/10.7554/eLife.68268.sa2

## Additional files

### Supplementary files
• Transparent reporting form

### Data availability
All data generated or analysed during this study are included in the manuscript and supporting files. Source data files have been provided for all the figures.

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
