## [Decision Letter]

**Acceptance summary:**

The insecticide Amitraz is used to control mite infestations in bee colonies. This is because bees are less susceptible to it than are mites. In this manuscript, the authors discover that three residues in the binding pocket of one of the mite octopamine receptors are what confer sensitivity to Amitraz in mites, and their variants in bees confer protection.

**Decision letter after peer review:**

Thank you for submitting your article "An octopamine receptor confers selective toxicity of amitraz on honeybees and Varroa mites" for consideration by *eLife*. Your article has been reviewed by 2 peer reviewers, including Sonia Sen as the Reviewing Editor and Reviewer #1 and overseen by K VijayRaghavan as the Senior Editor. The following individual involved in review of your submission has agreed to reveal their identity: Naoki Yamanaka (Reviewer #2).

Essential Revisions:

We think that this kind of approach to understanding the biology of the insecticides used in the field is interesting and important. With that in mind we are recommending the following essential revisions to conclusively demonstrate the mechanism of selective toxicity of Amitraz.

1) A comparison of the binding efficacies of all the bee and fly octopaminergic receptors to Amitraz in a heterologous system. Is it that only in mites Amitraz binds all 4 octopaminergic receptors?

2) If that's true, then, misexpressing the 4 mite receptors in fly neurons should make Amitraz toxic for flies.

*Reviewer #1:*

Bee colonies are often destroyed by Varroa mite infestations. To selectively target the mites while sparing the bees, the insecticide Amitraz is often used. In this manuscript, the authors address why bees are less susceptible to Amitraz, than are mites.

Amtiraz is known to act on Octopamine receptors. So, the authors identify octopamine receptors in the Varroa mites based on homology to *Drosophila*. They identify 4, which correspond to the 4 families in flies and show that they all four are activated by Amitraz and it's metabolic intermediary. Importantly, they show that the bee homologue of one of them (Octβ2R), is less sensitive to Amitraz. They perform homology modelling and identify 3 residues in the binding pocket of Octβ2R that is unique between bees and mites. They modify these 3 residues in the bee Octβ2R (in vitro) and show that these 3 residues do indeed confer sensitivity to Amitraz in vitro. These claims are well supported.

The authors next test this in vivo in *Drosophila* by expressing either the bee Octβ2R, the mite Octβ2R, or the 'mite-type bee Octβ2R' in all fly neurons and assaying two types of behaviours: aggression and activity. Flies expressing mite Octβ2R are more active and aggressive than those expressing the bee Octβ2R. However, flies expressing the 'mite-type bee Octβ2R' are behaviourally more similar to flies expressing the mite Octβ2R. These claims are also well supported.

Overall the data are clear and well presented and the manuscript clearly written. I wonder whether the insecticidal mode of action of Amitraz is via the behavioural changes that the authors model in flies or via a secondary mode of action that leads to toxicity.

My primary concern with this manuscript is whether the behaviour the authors model is what results in toxicity. I do not see a direct link to this in this version of the manuscript. This, along with some other concerns/suggestions I have, are listed below.

– Amitraz seems to bind effectively to all mite receptors. It likely binds to the fly OctB2R (with unknown efficacy), yet flies seem to tolerate Amitraz just fine. This makes me wonder if the selective toxicity of Amitraz mites is because it binds to all the other mite Oct receptors. The behavioural effects of Amitraz the authors have shown maybe shared between bees and flies.

– The Am, Dm, Vd, Rm OctB2R seem most closely related and it is the focus of this entire manuscript. I wonder if the authors want to focus on the ligand binding pocket of these 4 receptors and correlate them with their Amitraz responses. They have done this between Am and Vd, I wonder what this domain looks like in Dm. (I realise Rm is not possible to study in vitro.) If the authors are proposing that these residues determine resistance vs sensitivity to Amitraz, one would predict that the Dm OctB2R is more similar to Am OctB2R than to Vd OctB2R. Is this true?

– If true, bees, like flies, might display behavioural abnormalities that don't cause lethality, but might adversely affect their colonies. This would be important to know. (We are asking the authors to merely consider this possibility, and are not asking for these experiments in the revision of this manuscript.)

– Have the authors considered placing the *Drosophila* work at the end of their manuscript? The story might read better in the following order:

Figure1>Figure4>Figure2>Figure3>Figure5.

*Reviewer #2:*

In the present manuscript, Guo and others investigate molecular bases of the selectivity of an acaricide/insecticide named amitraz, which is used to control Varroa mites in apiculture. Parasitic mites are considered as one of the major reasons for the colony collapse in honeybees, so their investigation has a direct impact on the health of this one of the most important pollinator species for humans.

The authors are effectively using fly genetics approaches to demonstrate that one octopamine (OA) receptor (Octbeta2R) is the major mediator of in vivo effects of amitraz in fruit flies, which may also hold true in honeybees. The authors also successfully show the potential molecular basis of the different sensitivities of Octbeta2R in insects and mites. However, considering that amitraz can activate all the Varroa mite OA receptors in vitro (Figure 1), there are multiple other scenarios one can come up with to explain selectivity of this acaricide. For example, one of the other OA receptors (such as Octalpha2R) in Varroa mites may be expressed in some neurons where it is not expressed in insects, and that may confer the selective toxicity of this compound. It therefore seems that there is a logical gap between the presented data and their major claim that the high sensitivity of Varroa Octbeta2R indeed confers selective toxicity of amitraz for mites.

Overall, the paper nicely demonstrates a molecular basis of the high sensitivity of the mite Octbeta2R to amitraz. The authors' claim that this high sensitivity indeed confers selective toxicity of this acaricide needs further investigation.

1) In order to exclude alternative possibilities causing the selectivity of amitraz, manipulation of Octbeta2R gene in Varroa mites seems critical. Having said that, I definitely understand that genetic manipulations in Varroa mites are technically challenging. I therefore suggest the authors to at least change the title and revise the introduction and discussion accordingly.

2) The design of the rescue experiments presented in Figure 5 is confusing. It is hard to understand why pan-neuronal expression of the OA receptors is supposed to rescue the Octbeta2R mutant phenotype. I would imagine that elav-Gal4-mediated overexpression of OA receptors would make many unrelated neurons responsive to OA and amitraz, which makes it impossible to interpret the behavior assay results properly. The authors need to express these OA receptors only in Octbeta2R-expressing cells.

3) p. 8, line 153: The authors state that octbeta2R[f05679] is a null allele. I cannot find any evidence or author statement in indicated references suggesting that this is indeed a null allele. Please explain how this allele was confirmed to be null.

4) As a member of the fly community, please refer to this website: https://bdsc.indiana.edu/about/acknowledge.html when using resources from BDSC and acknowledge them accordingly with appropriate grant numbers so they can get continuous funding support.

---

## [Author Response]

Essential Revisions:We think that this kind of approach to understanding the biology of the insecticides used in the field is interesting and important. With that in mind we are recommending the following essential revisions to conclusively demonstrate the mechanism of selective toxicity of Amitraz.1) A comparison of the binding efficacies of all the bee and fly octopaminergic receptors to Amitraz in a heterologous system. Is it that only in mites Amitraz binds all 4 octopaminergic receptors?

To further explain the selectivity, we expressed the three remaining receptors from the fly and honeybee in HEK293 cells and found that all but DmOAMB showed responses to amitraz and DPMF in nanomolar concentrations (Figure 5—figure supplement 3). These results indicate that amitraz and DPMF can bind all four OA receptors in mites, flies, and honeybees with different efficacies.

2) If that's true, then, misexpressing the 4 mite receptors in fly neurons should make Amitraz toxic for flies.

Since the above hypothesis is not true, there is no need to do these experiments.

Reviewer #1:Bee colonies are often destroyed by Varroa mite infestations. To selectively target the mites while sparing the bees, the insecticide Amitraz is often used. In this manuscript, the authors address why bees are less susceptible to Amitraz, than are mites.Amtiraz is known to act on Octopamine receptors. So, the authors identify octopamine receptors in the Varroa mites based on homology to *Drosophila*. They identify 4, which correspond to the 4 families in flies and show that they all four are activated by Amitraz and it's metabolic intermediary. Importantly, they show that the bee homologue of one of them (Octβ2R), is less sensitive to Amitraz. They perform homology modelling and identify 3 residues in the binding pocket of Octβ2R that is unique between bees and mites. They modify these 3 residues in the bee Octβ2R (in vitro) and show that these 3 residues do indeed confer sensitivity to Amitraz in vitro. These claims are well supported.The authors next test this in vivo in *Drosophila* by expressing either the bee Octβ2R, the mite Octβ2R, or the 'mite-type bee Octβ2R' in all fly neurons and assaying two types of behaviours: aggression and activity. Flies expressing mite Octβ2R are more active and aggressive than those expressing the bee Octβ2R. However, flies expressing the 'mite-type bee Octβ2R' are behaviourally more similar to flies expressing the mite Octβ2R. These claims are also well supported.Overall the data are clear and well presented and the manuscript clearly written. I wonder whether the insecticidal mode of action of Amitraz is via the behavioural changes that the authors model in flies or via a secondary mode of action that leads to toxicity.My primary concern with this manuscript is whether the behaviour the authors model is what results in toxicity. I do not see a direct link to this in this version of the manuscript. This, along with some other concerns/suggestions I have, are listed below.

The insecticidal activity is very unusual for amitraz. At low doses, amitraz can induce hyperactivity and cause abnormal behaviors such as dispersal from plants and detachment of ticks from their host, presumably induced by higher motor activity. At higher doses, this hyperactivity can induce tremors that lead to death.

We have carried out new experiments by artificially activating *octβ2R* neurons using the thermosensitive cation channel *Drosophila* TRPA1. We saw that when *octβ2R* neurons were activated, animals showed strongly induced hyperactivity behavior and eventually led to paralysis (Video 1, Video 2), which is similar to the amitraz-induced behavior phenotype in insects and mites. However, activation of other OA receptor-expressing neurons did not show amitraz-induced poisoning symptoms (Figure 4). These results robustly demonstrate that pharmacological activation of Octβ2R by amitraz in vivo leads to toxicity and finally to death.

Amitraz seems to bind effectively to all mite receptors. It likely binds to the fly OctB2R (with unknown efficacy), yet flies seem to tolerate Amitraz just fine. This makes me wonder if the selective toxicity of Amitraz mites is because it binds to all the other mite Oct receptors. The behavioural effects of Amitraz the authors have shown maybe shared between bees and flies.

As we described in the reply to essential question 1, amitraz and DPMF can bind to all four OA receptors from mites, flies and honeybees with different efficacies. DmOctβ2R was 3-fold less sensitive to amitraz (EC_50_ = 242.1 nM) than VdOctβ2R, but 5-fold more sensitive to DPMF (EC_50_ = 1.4 nM) than VdOctβ2R (Figure 5A and 5B). We further found that flies can detoxify amitraz with its own metabolic enzymes, which is different from bees (Figure 5—figure supplement 2). A previous report found that the housefly can metabolize amitraz with CYP12A1, which is conserved in *Drosophila* genome but absent in the genomes of honeybees, *Varroa* mites and cattle ticks (Guzov et al., 1998, Arch Biochem Biophys 359:231–240).

The Am, Dm, Vd, Rm OctB2R seem most closely related and it is the focus of this entire manuscript. I wonder if the authors want to focus on the ligand binding pocket of these 4 receptors and correlate them with their Amitraz responses. They have done this between Am and Vd, I wonder what this domain looks like in Dm. (I realise Rm is not possible to study in vitro.) If the authors are proposing that these residues determine resistance vs sensitivity to Amitraz, one would predict that the Dm OctB2R is more similar to Am OctB2R than to Vd OctB2R. Is this true?

As we described in the above reply, fly and honeybee may employ different mechanisms to resist amitraz. We also aligned Octβ2R protein sequences from *Drosophila* and other species and found that the three predicted residues in the *Drosophila* amitraz binding site are different from those of bees. The two residues in extracellular loop 2 and TM6 are also different from Varroa’s (Figure 5F). Therefore, there is no need to correlate the sequences and the amitraz responses with DmOctβ2R.

If true, bees, like flies, might display behavioural abnormalities that don't cause lethality, but might adversely affect their colonies. This would be important to know. (We are asking the authors to merely consider this possibility, and are not asking for these experiments in the revision of this manuscript.)

Our new results indicate that the VdOctβ2R and DmOctβ2R are more sensitive to amitraz; however, AmOctβ2R is less sensitive to amitraz. Thus, we believe that amitraz at the recommended concentration in the field may have less adverse effects on honeybee colonies. Further research would be required to determine whether chronic exposure of amitraz affects honeybee behaviors negatively.

Have the authors considered placing the *Drosophila* work at the end of their manuscript? The story might read better in the following order:Figure1>Figure4>Figure2>Figure3>Figure5.

We think the *Drosophila* behavioral results (Figure 2 and Figure 3) are very important for identifying that Octβ2R as the sole molecular target of amitraz. Then we confirm that target-site insensitivity is the major mechanism for bees to resist amitraz.

Reviewer #2:In the present manuscript, Guo and others investigate molecular bases of the selectivity of an acaricide/insecticide named amitraz, which is used to control Varroa mites in apiculture. Parasitic mites are considered as one of the major reasons for the colony collapse in honeybees, so their investigation has a direct impact on the health of this one of the most important pollinator species for humans.The authors are effectively using fly genetics approaches to demonstrate that one octopamine (OA) receptor (Octbeta2R) is the major mediator of in vivo effects of amitraz in fruit flies, which may also hold true in honeybees. The authors also successfully show the potential molecular basis of the different sensitivities of Octbeta2R in insects and mites. However, considering that amitraz can activate all the Varroa mite OA receptors in vitro (Figure 1), there are multiple other scenarios one can come up with to explain selectivity of this acaricide. For example, one of the other OA receptors (such as Octalpha2R) in Varroa mites may be expressed in some neurons where it is not expressed in insects, and that may confer the selective toxicity of this compound. It therefore seems that there is a logical gap between the presented data and their major claim that the high sensitivity of Varroa Octbeta2R indeed confers selective toxicity of amitraz for mites.Overall, the paper nicely demonstrates a molecular basis of the high sensitivity of the mite Octbeta2R to amitraz. The authors' claim that this high sensitivity indeed confers selective toxicity of this acaricide needs further investigation.1) In order to exclude alternative possibilities causing the selectivity of amitraz, manipulation of Octbeta2R gene in Varroa mites seems critical. Having said that, I definitely understand that genetic manipulations in Varroa mites are technically challenging. I therefore suggest the authors to at least change the title and revise the introduction and discussion accordingly.

We revised the introduction, results, and discussion in the manuscript. We also performed new experiments to resolve the above concerns. We used the thermosensitive cation channel *Drosophila* TRPA1 to activate OA receptor-expressing neurons. We found that only the activation of *octβ2R* neurons mimics amitraz poisoning symptoms in target pests, while activation of others does not.

2) The design of the rescue experiments presented in Figure 5 is confusing. It is hard to understand why pan-neuronal expression of the OA receptors is supposed to rescue the Octbeta2R mutant phenotype. I would imagine that elav-Gal4-mediated overexpression of OA receptors would make many unrelated neurons responsive to OA and amitraz, which makes it impossible to interpret the behavior assay results properly. The authors need to express these OA receptors only in Octbeta2R-expressing cells.

Unfortunately, the only available *octβ2R-Gal4* strain is a knock-in allele in which a T2A-Gal4 coding sequences were introduced to the C terminus of *octβ2R*. Thus, we cannot use this strain to perform rescue assays in the *octβ2R* null mutant background. Anyway, the new thermogenetic assay data strongly suggest that activation of *octβ2R*-expressing neurons is sufficient to induce hyperactivity.

3) p. 8, line 153: The authors state that octbeta2R[f05679] is a null allele. I cannot find any evidence or author statement in indicated references suggesting that this is indeed a null allele. Please explain how this allele was confirmed to be null.

*octβ2R^f05679^* contains a piggyBac transposon inserted in the 5th exon of the octβ2R gene, interrupting the coding sequence. The female *octβ2R^f05679^* homozygous mutants are sterile, a phenotype consistence with octopamine-deficiency mutants (Lim et al., 2014, PLoS ONE 9(8): e104441). We added this references in the revised manuscript.

4) As a member of the fly community, please refer to this website: https://bdsc.indiana.edu/about/acknowledge.html when using resources from BDSC and acknowledge them accordingly with appropriate grant numbers so they can get continuous funding support.

We added this acknowledgement in the revised manuscript.